

# Two novel outlier detection approaches based on unsupervised possibilistic and fuzzy clustering

Zeynel Cebeci[1], Cagatay Cebeci[2], Yalcin Tahtali[3] and Lutfi Bayyurt[3]

[1] Department of Animal Science, Faculty of Agriculture, Cukurova University, Adana, Turkey
[2] Department of Electronics & Electrical Engineering, University of Strathclyde, Glasgow, United Kingdom
[3] Department of Agriculture, Faculty of Agriculture, Tokat Gaziosmanpasa University, Tokat, Turkey

## ABSTRACT

Outliers are data points that significantly deviate from other data points in a data set because of different mechanisms or unusual processes. Outlier detection is one of the intensively studied research topics for identification of novelties, frauds, anomalies, deviations or exceptions in addition to its use for data cleansing in data science. In this study, we propose two novel outlier detection approaches using the typicality degrees which are the partitioning result of unsupervised possibilistic clustering algorithms. The proposed approaches are based on finding the atypical data points below a predefined threshold value, a possibilistic level for evaluating a point as an outlier. The experiments on the synthetic and real data sets showed that the proposed approaches can be successfully used to detect outliers without considering the structure and distribution of the features in multidimensional data sets.

# INTRODUCTION

There are various definitions of the term "outlier" in statistics and data science. According to *Barnett & Lewis (2004)*, an outlier is "an observation (or subset of observations) which appears to be inconsistent with the remainder of that set of data" as quoted in *Hodge & Austin, (2004)*. Outliers are the data points that violate the assumptions with the statistical data distribution models. These data points behave differently than the others in the same data set, and hence, lead to suspicion as if they are created by different factors or exceptional processes. The outliers should be removed from data sets before going to further data analysis to avoid the biased results in the calculation of descriptive statistics and other statistical model parameters used in many data mining applications. For this reason, outlier detection for identifying outliers in data sets is an important data preprocessing task in statistics and data science. Besides its use for this sort of data cleansing purposes, outliers also provide valuable information in discovering unusual objects, unexpected situations, fraudulent behaviours, novel cases, rare events, human errors and device faults. Therefore, outlier detection is used for detecting novelties, frauds, anomalies, intrusions,

Corresponding author
Yalcin Tahtali,
yalcin.tahtali@gop.edu.tr

noises, deviations, faults or exceptions in business and engineering applications (*Hodge & Austin, 2004*).

According to the surveys, various methods and approaches have been proposed for outlier detection (*Patcha & Park, 2007*; *Chandola, Banerjee & Kumar, 2009*; *Gogoi et al., 2011*; *Zhang, 2013*; *Kalinichenko, Shanin & Taraban, 2014*). All these methods and approaches can be grouped in different taxonomies such as univariate vs multivariate; parametric, semi-parametric vs non-parametric; supervised, semi-supervised vs unsupervised; or more frequently in the classes of distribution-based, depth-based, distance-based, density-based and clustering-based methods (*Ben-Gal, 2005*; *Cebeci, 2020*; *Zhang, Meratnia & Havinga, 2010*).

Statistical outlier detection methods (a.k.a. distribution-based methods) suppose that outliers are the data objects having a low probability of belonging to a modelled distribution, and therefore, deviate from the assumptions of the concerned distribution. In general, they are based on statistical tests, and their efficiencies highly depend on whether the assumptions of statistical models fit to the real data or not. These methods generally cannot be applied on multidimensional data sets, and they also do not work for data sets in which there is no prior information about their distribution (*Ben-Gal, 2005*; *Cebeci, 2020*).

The proximities of data points in a data set can be obtained using the distance or density metrics. The distance-based proximity methods mark a data point as an outlier if it is farther away from other data points in the same data set. On the other hand, density-based proximity methods treat a data point as an outlier if it is located in a low-density region in a data set. In outlier detection, the overall performance of a proximity-based method can be easily affected by the metrics used with it (*Cebeci, 2020*). The advantage of these methods over statistical-based methods is that no prior assumptions are made about data distribution. However, these methods have computational complexities and sometimes lead to difficulties in identifying outliers when they are very close to normal data points.

The clustering-based methods seek the data points that are very distant to the closest cluster centers. These methods suppose that normal data points belong to big and point-intensive clusters, whereas outlying data points are not located in any cluster or belong to very small clusters. Some people consider cluster-based outlier detection as computationally expensive and not scalable for large data sets. Although this is true to some extent, as concluded in many studies, there is *n* superior outlier detection method for universally applicable on all kind of data sets. The clustering-based methods can be used in outlier detection practically because they need not prior information about the structure and distribution of data. Furthermore, they are still good candidates for outlier detection in multidimensional data sets and provide advantages to work in incremental mode. Sreevidya argued that the clustering-based outlier detection was more successful than the other types of outlier detection methods with an 88% accuracy rate in an experiment with many data sets (*Sreevidya, 2014*). Similarly, *Christy, Gandhi & Vaithyasubramanian (2015)* found that their clustering-based outlier detection approach provided better performance than the other distance-based outlier detection algorithms. In a very recent comparative study, *Goldstein & Uchida (2016)* argued that outlier detection based on clustering can be a good option for working on large data sets.

A dozen of fuzzy and possibilistic clustering algorithms have been developed in the last two decades. Although they mainly target to identify homogeneous groups in data sets, their possibilistic partitioning results can be also used in detecting outliers. Moreover, some existing clustering algorithms have also been extended to improve outlier detection. Because of their less sensitivity to noise and outliers, the fuzzy and possibilistic clustering algorithms are expected to be more efficient in detecting outliers when compared to the conventional hard and soft partitioning algorithms. In this study, we describe two simple approaches to detect outliers in multidimensional data sets using the possibilistic partitioning results from the possibilistic and fuzzy clustering algorithms.

## RELATED WORKS

### Studies on clustering-based outlier detection

Choosing an appropriate clustering algorithm for outlier detection is difficult and usually application dependent task because it depends on many criteria such as the size of data set, types, distribution and number of features, level of outliers in data, shapes of existing clusters, time constraints and many others (*Ben-Gal, 2005*; *Penny & Jolliffe, 2001*). Although there is a few outlier detection techniques based on the hierarchical clustering algorithms (*Loureiro, Torgo & Soares, 2004*), the partitioning algorithms have been widely used in outlier detection especially when the size and number of features are primary concerns. Among them, K-means has been the most widely used algorithm in many clustering-based outlier detection studies. On the other hand, as mentioned in many studies, K-means is sensitive to noise and outliers, and may not give accurate results (*Hodge & Austin, 2004*; *Gan & Ng, 2017*). Alternatively, K-medoids or partition around medoids (PAM) is less sensitive to local minima problem and, therefore, some studies targeted to use these hard clustering algorithms in outlier detection (*Jayakumar & Thomas, 2013*; *Kumar, Kumar & Singh, 2013*). However, the hard clustering algorithms such as K-means and PAM force each data point to belong to the nearest cluster. In this case, it becomes difficult to find outliers when they do not tend to form small or sparse clusters.

As a soft clustering algorithm, the Fuzzy C-means (FCM) algorithm calculates different membership degrees for each data point to every cluster. When a data point has the same degree of membership to the clusters in a partitioning task it can be evaluated as an outlier. For detection of outliers using fuzzy clustering algorithms, *Klawonn & Rehm (2005)* proposed a method that combines FCM with a modified version of *Grubbs (1969)*, which is a statistical outlier detection test for univariate data. In their approach, the mean and standard deviation of each feature for each cluster is calculated after a clustering analysis, and then each feature is tested against a critical value. The feature vector with the largest distance to the mean vector is assumed to be an outlier, and removed from the data set. With the new data sets, the outlier tests are repeated until no outlier is found. The other clusters are processed in the same way.

*Rehm, Klawonn & Kruse (2007)* introduced an outlier detection method using noise clustering with FCM. Their approach determines the noise distance preserving the hyper-volume of the feature space when approximating the feature space using a specified number

of prototype vectors. They obtained high accuracy with their approach for different numbers of clusters in FCM runs.

*Moh'd Belal, Al-Dahoud & Yahya (2010)* also used the FCM algorithm and proposed a method based on testing the difference of the objective function values. In their method, small clusters are determined as outlier clusters and removed from data set after the first run of FCM. The remaining outliers are then determined by computing the differences between objective function values by temporally removing the points from the analyzed data set.

Noise may be present in data sets due to random errors in a feature and they should not be considered as outliers. But they may distort the distribution and mask the distinction between normal objects and outliers. Regarding this kind of problems, the possibilistic algorithms are good candidates for distinguishing outliers more efficiently. For example, *Treerattanapitak & Jaruskulchai (2011)* stated that integrating the possibilistic and fuzzy terms in a clustering algorithm allows detecting outliers. In their study, possibilistic exponential fuzzy clustering produced accurate results in outlier detection based on exponential outlier factor scores that are calculated from the distances to the centroids.

Even though the main goal of clustering algorithms is to divide data set into homogenous clusters, in recent years, there is an increasing interest to extend them with some approaches for detecting outliers. *Duan et al. (2009)* proposed the CBOF, a clustering-based outlier detection algorithm requires four parameters. *Huang et al. (2016)* introduced the NOF algorithm, a non-parameter based on natural neighbor. *Gan & Ng (2017)* shortly reviewed the extended algorithms based on K-means and introduced the K-means with outlier removal (KMOR) algorithm which handles clusters and outliers simultaneously. Recently, *Huang et al. (2017)* proposed the ROCF, a cluster detection algorithm does not require a top-$n$ parameter. ROCF detects isolated outliers and outlier clusters based upon a k-NN graph. The idea behind ROCF is that outlier clusters are smaller in size than normal clusters.

Although all the algorithms discussed above have their own advantages, they need to select at least one or more parameters. So, using the simple methods which do not require data-dependent parameters may be more useful in outlier detection. Although the typicality degrees from an optimal run of a possibilistic clustering algorithm can be used to identify outlying points, since the fuzzy and possibilistic algorithms are more robust to the noise and coincident clusters problems they can be more successful to detect the outliers. In this study, we used the unsupervised fuzzy possibilistic clustering algorithm by *Wu et al. (2010)* as the representative of fuzzy and possibilistic algorithms.

## Unsupervised fuzzy and possibilistic clustering algorithms

The hard prototype-based clustering algorithms, *i.e.*, K-means and its variants, assume that each object belongs to only one cluster; however, clusters may overlap and objects may belong to more than one cluster. In this case, a data point can be a member of several clusters with varying membership degrees between zero and one. The Fuzzy C-means (FCM) clustering algorithm (*Bezdek, 1993*) assigns a fuzzy membership degree to each data point based on their distances to the cluster centers. If a data point is closer to a cluster

center, its membership degree to that cluster will be higher than its membership degrees to the other clusters.

To fix the sensitivity of FCM to noise and outliers, the possibilistic C-means (PCM) algorithm has been introduced (*Krishnapuram & Keller, 1993*). However, it has been revealed that PCM can generate coincident clusters if it is not well initialized. Later a dozen of hybrid versions of FCM and PCM have been proposed to overcome the problems with FCM and PCM. For instance, the fuzzy possibilistic C-means (FPCM) *Pal, Pal & Bezdek (1997)* aims to compute memberships and typicalities simultaneously. An extended version of PCM has been introduced by *Timm et al. (2001)* to make clusters far away from each other. The possibilistic fuzzy C-means (PFCM) *Pal et al. (2005)* has also been developed to fix the noise sensitivity problem with FCM, the coincident clusters problem with PCM and row sum constraints problem with FPCM.

*Wu et al. (2010)* proposed the unsupervised possibilistic and fuzzy clustering (UPFC) algorithm as an improved version of the original possibilistic clustering algorithm (PCA) (*Yang & Wu, 2006*). UPFC combines FCM and PCA to overcome the noise sensitivity problem of FCM and the coincident clusters generated by PCA. Moreover, unlike PCA algorithm, UPFC also does not require a fuzzy membership matrix from a previous FCM run. Using the objective function in Eq. (1), UPFC minimizes the distances between $c$ prototype vectors ($\boldsymbol{v}_i$) and $n$ feature vectors ($\boldsymbol{x}_k$) in $R^p$ features space.

$$J_{UPFC}(\boldsymbol{X};\boldsymbol{U},\boldsymbol{V}) = \sum_{k=1}^{n}\sum_{i=1}^{c}(au_{ik}^m + t_{ik}^\eta)d^2(\boldsymbol{x}_k,\boldsymbol{v}_i) + \frac{\beta}{n^2\sqrt{c}}\sum_{k=1}^{n}\sum_{i=1}^{c}(t_{ik}^\eta log t_{ik}^\eta - t_{ik}^\eta) \qquad (1)$$

In Eq. (1), $u_{ik}$ and $t_{ik}$ are the fuzzy and possibilistic membership degree of $k^{th}$ feature vector ($\boldsymbol{x}_k$) to the $i^{th}$ cluster respectively. The UPFC objective function has the constraints which are listed in Eq. (2).

$$\sum_{i=1}^{c}u_{ik} = 1; \forall k; 0 \le u_{ik} \le 1; a > 0; b > 0; m > 1; \eta > 1. \qquad (2)$$

In the possibilistic algorithms, the parameter $m$ is a fuzziness exponent and the parameter $\eta$ is a typicality exponent. Both these parameters are usually set to 2. In the objective function in Eq. (1), the parameters $a$ and $b$ are weighing coefficients for setting the relative importance of fuzziness and typicality respectively. Generally, both of these coefficients are equally set to 1.

To minimize $J_{UPFC}$, the typicality degrees $t_{ik}$ and the membership degrees $u_{ik}$ are recalculated in each iteration step with Eqs. (3) and (4) respectively.

$$u_{ik} = \left(\sum_{j=1}^{c}\left(\frac{d(\boldsymbol{x}_k,\boldsymbol{v}_i)}{d(\boldsymbol{x}_k,\boldsymbol{v}_j)}\right)^{2/(m-1)}\right)^{-1} \forall i,k \qquad (3)$$

$$t_{ik} = exp\left(\frac{bn\sqrt{c}d^2(\boldsymbol{x}_k,\boldsymbol{v}_i)}{\beta}\right)\forall i. \qquad (4)$$

As seen from the formula in Eq. (4), a typicality degree is a possibilistic measure indicating the membership degree of a data point to clusters. The value $\beta$ is is distance variance, and calculated as seen in Eq. (5).

$$\beta = \frac{1}{n}\sum_{k=1}^{n}d^2(\boldsymbol{x}_k, \overline{x}); \overline{x} = \frac{1}{n}\sum_{k=1}^{n}\boldsymbol{x}_k \tag{5}$$

The formula in Eq. (6) is used to update the cluster centers in each iteration. The UPFC algorithm stops when a predefined convergence level is achieved.

$$v_i = \frac{\sum_{k=1}^{n}(au_{ik}^m + t_{ik}^\eta)\boldsymbol{x}_k}{\sum_{k=1}^{n}(au_{ik}^m + t_{ik}^\eta)} \forall i \tag{6}$$

## PROPOSED APPROACHES FOR OUTLIER DETECTION

To find the outliers in multidimensional data sets, we describe the proposed novel approaches in this section. The pseudocode for the proposed approaches is given in Algorithm 1. This algorithm uses the matrix of possibilistic membership degrees (a.k.a typicality degrees). A matrix of possibilistic membership degrees is returned from a UPFC run and passed to the proposed algorithm to find the outliers in the studied data set. In the subsections below the proposed approaches are explained.

### Approach 1

While the fuzzy clustering algorithms determine the fuzzy membership degree of fuzzy data points to any cluster, they do not evaluate typicality according to their distances from cluster centers. For example, suppose there are two data points A and B, both they have 50% fuzzy membership degree to two different clusters. If their distances from the cluster centers are, say, 2r and 5r, the latter will be a more atypical data point as it is further away from the cluster centers. When compared to fuzzy membership degrees, typicality degrees from the fuzzy and possibilistic clustering algorithms reveal the distinction between the highly atypical and the less atypical members of the clusters. This means that all fuzzy points for a cluster are not equivalent: some are more typical and some are not. This implies that typicality is distinct from a simple similarity to the cluster center because it also involves a dissimilarity notion. Therefore, in our proposed algorithms we used the typicality results from UPFC algorithm.

In our first approach, a data point which is not a member of any cluster is evaluated as an outlier. In such case, its average typicality to all clusters should be less than a predefined possibilistic threshold level ($\alpha$). This approach checks whether the average typicality of $k$th feature vector ($x_k$) to all clusters exceeds a threshold level, which is a user-defined possibility degree for evaluating a data point as an outlier.

The test function in Eq. (7) is used to label a data point as an outlying data point. If the average typicality of $x_k$ to all clusters ($c$ clusters) is less than $\alpha$, it is considered as a highly atypical data point and flagged as an outlier in the data set (see in Algorithm 1).

**Algorithm 1 Pseudocode for the proposed outliers detection approaches.**

```
1    Input: T, alpha, apr
2    //Typicality degrees matrix in nxc dimension, and built by an
3    //unsupervised possibilistic clustering algorithm
4    // alpha, threshold possibility value for outlier testing
5    // apr, number of the approach to be used in outlier detection
6    Output: Outliers
7    //Outliers, vector of n length to store the flags of outliers
8    n <- count of rows of matrix T
9    c <- count ofcolumns of matrix T
10   // If alpha is undefined, use 0.05 as the default value
11   if alpha is null then alpha = 0.05
12   Outliers <- {0} //Assign 0 to all elements of the outliers
13   for k = 1 to n do
14      if apr = 1 then
15         sumT <- 0
16         for i = 1 to c do
17            sumT <- sumT + T[i,k]
18         end
19         avgT <- sumT / c
20         if avgT <= alpha then
21            Outliers[k] <- 1
22         end
23      else
24         if apr = 2 then
25            isOutlier <- True
26            for i = 1 to c do
27               if T[i,k] >= alpha then
28                  isOutlier <- False
29               end
30            end
31            if isOutlier = True then
32               Outliers[k] <- 1
33            end
34         end
35      end
36   end
37   return Outliers
```

Choosing an appropriate $\alpha$ value is crucial in this approach. According to our tests on several experimental data sets, we determined that a threshold level of 1% ($\alpha = 0.01$) may be sufficient to find the outliers on the results of UPFC runs, and therefore we recommend

to use it as the default value for approach on large data sets.

$$is.outlier(x_k) = \begin{cases} 1 & if \left(\sum_{i=1}^{c} t_{ik}/c\right) \leq \alpha \\ 0 & otherwise \end{cases} \tag{7}$$

## Approach 2

In the second approach, it is assumed that a data point is an outlier if it is atypical for every cluster in ad a data set. As seen in Eq. (8), if the typicality of the feature vector $x_k$ for all clusters is less than a user-defined threshold level ($\alpha$), it is evaluated as an outlier, otherwise, it is normal (see in Algorithm 1). As expected with this function, larger threshold values may lead to less number of outliers whereas smaller ones may lead to much more number of outliers. However, we recommend to use a default threshold typicality level of 5% ($\alpha = 0.05$) for finding the most probable outliers in data sets, one can increase it to higher levels to make the detection task looser. But, in this case, the probability of treating normal data points as outliers (false positives) also increase. In our experiments, a threshold level up to 10% was sufficient to detect the simulated outliers in many runs of UPFC on several synthetic data sets. One can look for an appropriate threshold with trial and error approach for different data sets or can use a prior known level of threshold for a specific application domain.

$$is.outlier(\boldsymbol{x}_k) = \begin{cases} 1 & if \ t_{ik} \leq \alpha; \forall i \\ 0 & otherwise \end{cases} \tag{8}$$

In some cases, even so, the proposed approaches can find most of the outliers in a data set, collective outliers do exist in it, and their presence should be checked. These are the outlying data points in small or sparse clusters. Thus, all members of small clusters could be seen as the collective outliers according to the third assumption is given in *Chandola, Banerjee & Kumar (2009)*. If the number of members of a cluster is less than a threshold cluster size it can be flagged as a small cluster. Several formulas have been presented to calculate a threshold size for determining small clusters. For example, *Loureiro, Torgo & Soares (2004)* stated that a cluster is small if its size less than half of the average size of $c$ clusters ($n/2c$). *Santos-Pereira & Pires (2002)* proposed that clusters whose size of $2p + 2$ can be considered as small outlier clusters. However, the mentioned proposals might be useful for small data sets they may not work well for larger and high dimensional data sets. For computing a more suitable threshold size to determine small clusters ($tcs$), we propose to use the formula in Eq. (9). This is a simple formula based on the expected cluster size, which is weighted with the logarithm of the number of features in an examined data set.

$$tcs = \frac{\log 2n}{c} \log 2p \tag{9}$$

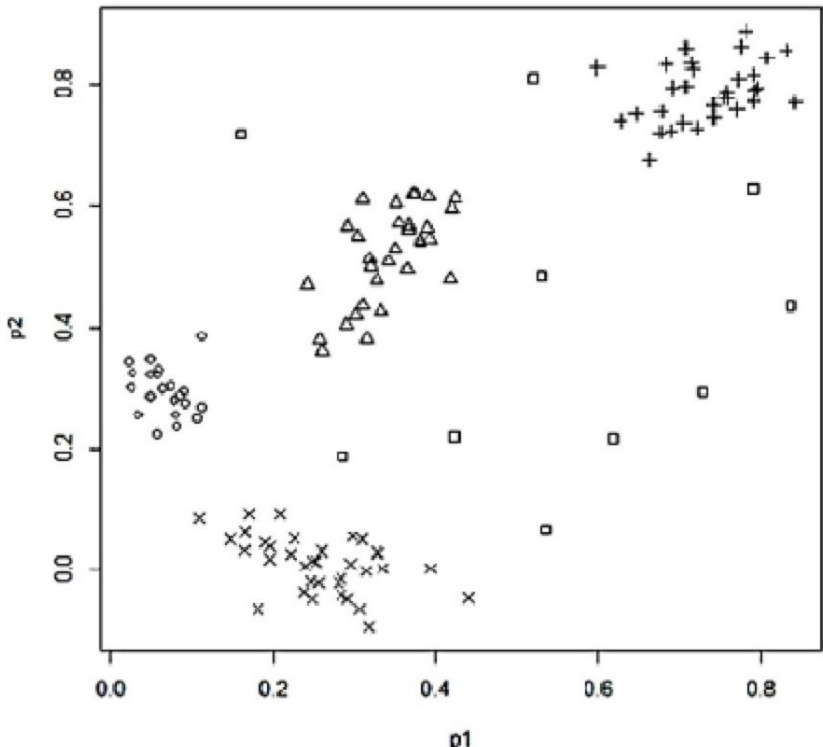

**Figure 1  2D scatter plot for SD1.**

## EXPERIMENTS ON SYNTHETIC DATA SETS

In this study, the proposed approaches have been tested on two synthetic data sets, one with three features ($p = 3$) and the other with two features ($p = 2$) for easy demonstration of outlier detection. We generated the first experimental data set by using the R package 'MixSim' (*R Core Team, 2020*). To examine the performance of the proposed approaches on multidimensional data, we aimed to generate our dataset consisting of three features and four clusters. For this purpose, we ran 'MixSim' with the set of parameters of *BarOmega* $= 0.001$, $K = 4$, and $p = 3$. The created data set had a total of 120 data points, 30 in each cluster. In addition to the normal data points located in the simulated clusters, 10 outlying data points were also added to the data set as illustrated with the box dots in Fig. 1. The labels (row numbers) of these outliers were between 121 and 130. As a result of the simulation work using 'MixSim', we obtained our first synthetic data set (SDS1) with 130 data points as illustrated with 2D and 3D scatter plots in Fig. 1.

In our experiments, we coded the R functions for the proposed outlier detection approaches in R environment (*R Core Team, 2022*). As an implementation of the UPFC algorithm, we used the 'upfc' function of the R package 'ppclust' (*Cebeci, 2019*). To initialize the cluster prototypes, we applied Kmeans++ algorithm *Arthur & Vassilvitskii (2007)* by using 'kmpp' function in the R package 'inaparc'. For initialization of cluster prototypes

matrix $V$ and membership matrix $U$, we set the seed of random number generator to a predefined constant number (234) to ensure the same initialization values between UPFC runs. To determine the success of proposed outlier detection approaches at different number of clusters, UPFC has been started at five levels of number of clusters ($c = 2, \ldots, 6$) on the synthetic data sets. An important subtask in clustering-based outlier detection is to find a good partitioning result for identifying outliers. A reasonable way to find an optimum partitioning is to validate the results obtained at different number of clusters. For this purpose, we used four well-known internal fuzzy indices: Xie-Beni index (XB) *Xie & Beni (1991)*, Tang-Sun index (TS) *Tang, Sun & Sun (2005)*, Pakhira-Bandyopadhyay-Maulik Fuzzy index (PBMF) *Pakhira, Bandyopadhyay & Maulik (2004)*, Modified Partition Coefficient (MPC) (*Dave, 1996*). These internal validation indexes work with membership degrees, which are calculated by the regular FCM algorithm and some of its newer variants. However, the fuzzy and possibilistic algorithms do not calculate fuzzy membership degrees only, they also calculate possibilistic membership degrees called as typicality degrees. Since the possibilistic clustering algorithms do not constraint row sums of feature vectors, the validation indices for fuzzy clustering cannot be directly used to validate typicality degrees in possibilistic and fuzzy environments. In this case, the existing indices do not work properly with typicality degrees. For instance, the MPC index value also becomes higher when the number of clusters used by a partitioning algorithm is higher. The same validation issues apply to other indexes. Thus, some modified or generalized versions of these indices are needed to find a good partitioning result by using typicality degrees. As a solution for this need, *Yang & Wu (2006)* proposed to normalize typicality degrees as formulated in Eq. (10).

$$u'_{ik} = \frac{t_{ik}}{\sum_{i=1}^{c} t_{ik}}; \forall i, k \tag{10}$$

Using Eq. (10), we divided the typicality degrees into their row sums and obtained the normalized typicality degrees ($u'_{ik}$) to use in validation of clustering results. In this study, the R package 'fcvalid', *Cebeci (2020)* has been run for computing the internal validity indices using the UPFC clustering results.

The outliers detected from the results of UPFC runs for five different number of clusters for SDS1 are given in Table 1.

As seen in Table 1, all the ten simulated outlying points in SDS1 plus one more data point (the point 39) were found as outliers by Approach 1 using the typicalities from UPFC run for the clustering done for two clusters ($c = 2$). Except for two points (the points 126 and 128), Approach 2 also detected most of the simulated outliers in SDS1. Using the results from partitioning done for three clusters ($c = 3$), Approach 1 found twelve outliers that consist of points 1 and 6 in addition to all of the simulated outliers. In the clustering done for three clusters, except two points (the points 124 and 128) Approach 2 also detected the same outliers as those found in the clustering done for two clusters.

For comparison purposes, as the second data set (SDS2), we used the synthetic data set given in Table 2 by *Rehm, Klawonn & Kruse (2007)*. The data set SDS2 contains 45 data points in two clusters with some amount of outliers. The labels of the outliers are 14, 35,

**Table 1  Outliers detected in SDS1.**

| c | Approach 1 | Approach 2 |
|---|---|---|
| 2 | 39 121 122 123 124 125 126 127 128 129 130 | 121 122 123 124 125 127 129 130 |
| 3 | 1 6 121 122 123 124 125 126 127 128 129 130 | 121 122 123 125 127 128 129 130 |
| 4 | 121 122 123 124 125 126 127 128 129 130 | 121 122 123 125 126 127 128 129 130 |
| 5 | 121 122 123 124 125 126 127 128 (129 130) | 121 123 125 126 128 |
| 6 | 104 121 122 123 124 125 126 127 128 | 121 123 125 126 128 |

**Table 2  Values of the validity indices by different number of clusters for SDS1.**

| c | XB | TS | PBMF | MPC |
|---|---|---|---|---|
| 2 | 0.1167624 | 8.043394 | 0.09970858 | 0.8583530 |
| 3 | 0.1385422 | 8.044566 | 1.87827975 | 0.8379177 |
| 4 | 0.1853725 | 8.732699 | 0.01513389 | 0.8392733 |
| 5 | 0.1059272 | 6.024347 | 0.95398914 | 0.8742973 |
| 6 | 1.4147905 | 10.132927 | 0.06037724 | 0.7632591 |

41, 42, 43, 44 and 45 (marked with the box dots in Fig. 2) according to the results obtained by *Rehm, Klawonn & Kruse (2007)*.

Although the proposed approaches were highly successful to identify the simulated outliers from the clustering results at $c = 2$ and $c = 3$, outlier detection should be based on an optimum clustering result suggested by the fuzzy validation indices. According to the validation indices in Table 2, the PBMF index suggested four clusters whereas XB, TS and PBMF suggested five clusters for SDS1. In UPFC run for four clusters, Approach 1 completely detected the ten simulated outliers in SDS1. Approach 2 was also successful and found the same result with Approach 1 excluding one missing outlier (the point 124).

The clusters and the detected outliers are also shown in Fig. 3 for giving a clear idea about the clusters and outliers results found with UPFC run for four and five clusters. The scatter plots on the left panels show the outliers found by Approach 1 while the scatter plots on the right panels stand for the results found by Approach 2. In Fig. 3, the blue crossed circles and the red box dots with labels show the cluster centroids and detected outliers, respectively. In the UPFC run for five clusters, the number of outliers was less than those detected from UPFC run for four clusters. As seen from the left two panels of Fig. 3, while the data points 129 and 130 are the outliers according to the results from UPFC run for four clusters, they moved to a newly formed cluster in UPFC run for five clusters. Since their typicalities to this newly formed cluster increased they were not detected as the outliers by the proposed approaches in UPFC run for five clusters. In such cases, the size of clusters should be examined for the existence of collective outliers because the small clusters are considered as the outlier clusters which having a few numbers of highly similar points in isolated locations of data space. In our experiment, the above-mentioned cluster includes the points 129 and 130 were interpreted as an outlier cluster because its size was

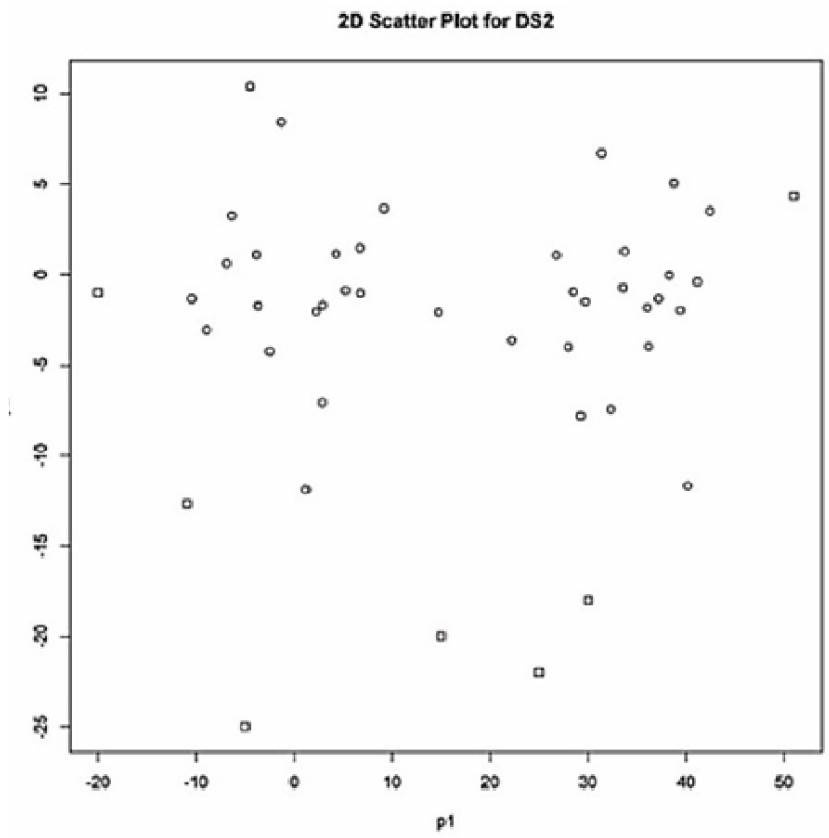

**Figure 2** 2D-scatter plot (p1 vs p2) for SDS2.

small according to the formula in Eq. (9). In Fig. 3, we can conclude that UPFC run for five clusters was also successful to detect all the simulated outliers in SDS1.

According to the results in the last row of Table 1, Approach 1 again detected the identical outliers plus one more (the point 104) in the clustering done for six clusters whereas Approach 2 found five outliers. These findings pointed out that some of the outliers may not be found directly when the number of cluster parameter used in the clustering algorithm is higher than the real number of groups in the examined data sets. In such case, as above discussed for the clustering done for five clusters, the size of clusters should be examined for finding the small clusters. In our experiment, since the points 129 and 130 were together located in a small cluster, they were again considered as the outliers in clustering done for six clusters.

As seen in Table 3, all the validity indices found that the best partitioning for SDS2 from the clustering done for two clusters ($c = 2$). As seen in Fig. 4, based on this validation result, four outliers (the points 42, 43, 44 and 45) and three outliers (the points 43, 44 and 45) were detected by the Approach 1 and Approach 2, respectively. The outliers detected by Approach 1 were identical between the clustering done for two and three clusters. But less number of outliers was detected with the clustering done for four clusters ($c = 4$). The

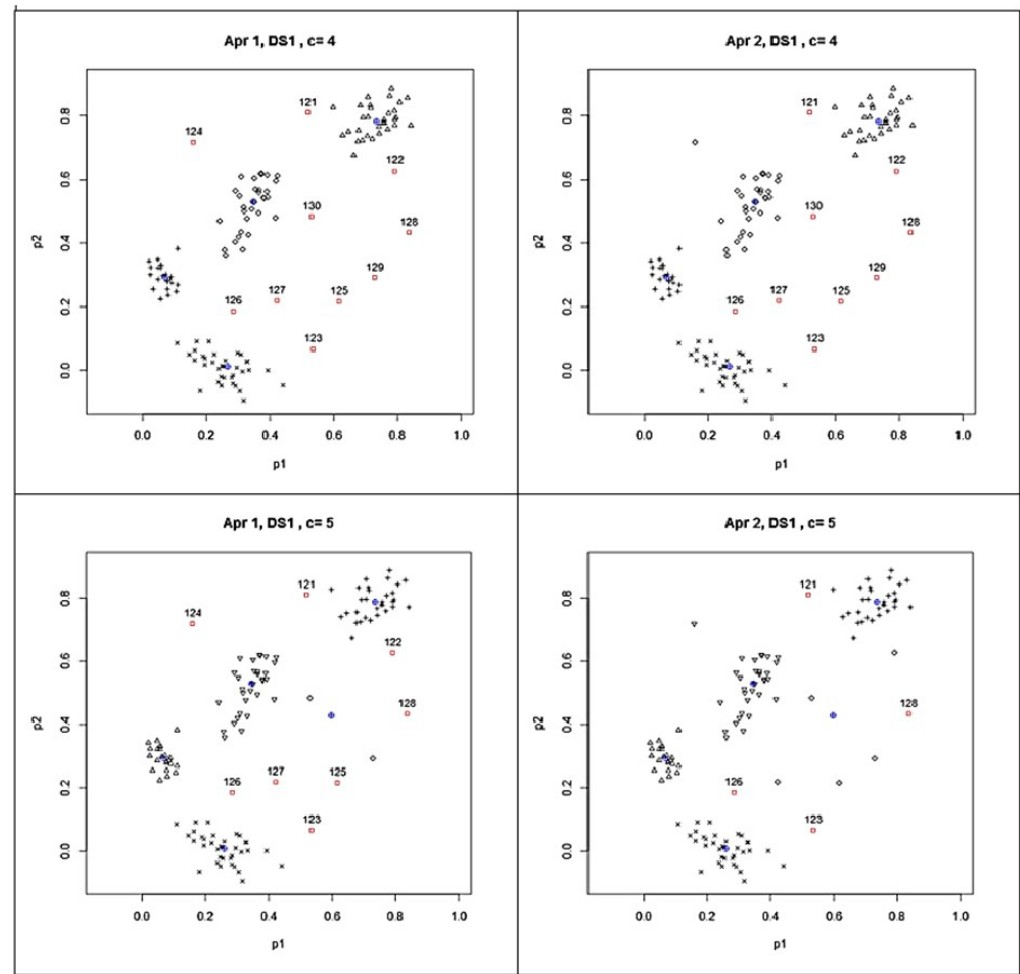

**Figure 3** Outliers detected from the results of possibilistic partitioning for four and five clusters on SDS1.

**Table 3** Values of the validity indices by different number of clusters for SDS2.

| c | XB | TS | PBMF | MPC |
|---|---|---|---|---|
| 2 | 0.08256631 | 4.213788 | 257.2904 | 0.9480826 |
| 3 | 0.85765530 | 43.048094 | 30203.7867 | 0.6795931 |
| 4 | 0.42083833 | 22.494335 | 42228.4537 | 0.7041015 |
| 5 | 3.36639018 | 177.540583 | 70887.5026 | 0.4275838 |
| 6 | 2.65760351 | 146.300871 | 45277.8600 | 0.4881886 |

outliers found with Approach 1 from the clustering done for five clusters ($c = 5$) were the same with those found in the study on SDS2 data set by *Rehm, Klawonn & Kruse (2007)*.

Similar to those found for SDS1, Approach 2 detected less number of outliers than Approach 1 for SDS2 too. However, this result was due to the used threshold level ($\alpha$) as small as 0.05. When it is increased to a higher level, namely to 2 $\alpha$, Approach 2 could

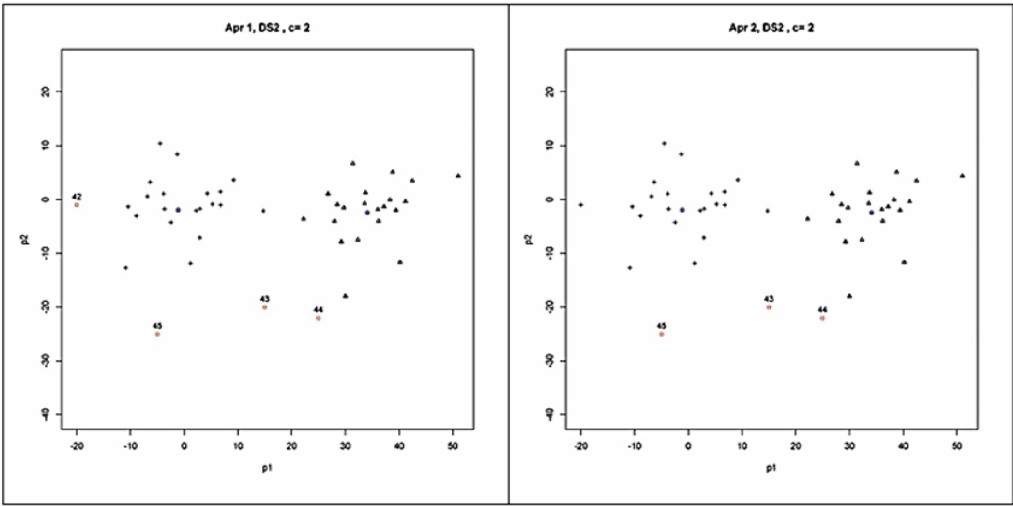

**Figure 4** Clusters and outliers detected from the result of possibilistic partitioning for two clusters on SDS2.

produce the same results with Approach 1. Even though leveraging $\alpha$ to higher values is an option for detecting more outliers, Approach 2 should be applied at lower levels of $\alpha$ to obtain the most possible outliers.

## COMPARISON OF MULTIVARIATE OUTLIER DETECTION METHODS

For comparison purposes, we have finally detected the outliers in SDS1 and SDS2 by using some outlier detection packages in R environment. The function *lofactor* in the package *DMwR* (*Torgo, 2010*) and the function *lof* in the package *Rlof* (*Hu et al., 2020*) calculate local outlier factors using the LOF (local outlier factor) algorithm with the parameter $k$, which is the number of neighbors used in the calculation of the local outlier factors. LOF is an algorithm for identifying density-based local outliers (*Breunig et al., 2000*). It compares the local density of a point with those of its neighbors. If the density is significantly lower than the density of its neighbors it is evaluated as an outlier.

The function *aq.plot* in *mvoutlier* package (*Filzmoser & Gschwandtner, 2021*) plots the ordered squared robust Mahalanobis distances of the observations against the empirical distribution function of the $MD_i^2$. The distance calculations are based on the MCD estimator. For outlier detection two different methods are used. If an observation exceeds a certain quantile of the Chi-squared distribution it is marked as an outlier by the first method. The second is an adaptive procedure searching for outliers specifically in the tails of the distribution, beginning at a certain Chi-squared quantile (*Filzmoser et al., 2005*). The function behaves differently depending on the dimension of the data. If the data is more than two-dimensional it is projected on the first two robust principal components (*Filzmoser & Gschwandtner, 2021*).

*HDoutliers* package (*Fraley, 2022*) is an R implementation of an outlier detection algorithm based on a distributional model that uses probabilities to determine outliers in multidimensional data sets. The details of the algorithm can be seen in *Wilkinson (2018)*. The first stage with the *HDoutliers* is to run *getHDmembers* function in which the data is partitioned according to exemplars and their associated lists of members. Afterward, an exponential distribution is fitted to the upper tail of the nearest-neighbor distances between exemplars with *getHDOutliers* function of the package. Data points are considered outliers if they fall in the $(1 - \alpha)$ tail of the fitted cumulative distribution function (*Fraley, 2022*).

As seen in Table 4, while the outliers detected in SDS1 with *HDOutliers* algorithm were the same with those found by Approach 1 and Approach 2 in the clustering done with four clusters in Table 2. *HDoutliers* did not outcome any outlier for SDS2. LOF algorithm, ran with the parameter $k = 10$ in *DMwR* and *Rlof* package produced the identical outliers (excluding the point 20) to those found by Approach 1 in the clustering for six clusters in SDS1. Although LOF algorithm detected more outliers when compared to those detected by the proposed approaches in SDS2, most of outliers found were the same. The *mvoutlier* package resulted with too many outliers for SDS1 (45 outliers), and thus, was not comparable to the other algorithms. But it suggested mostly similar outliers to those of Approach 1 in SDS2. Finally the outliers found by Approach 1 from the clustering done for five clusters ($c = 5$) were the same with those found by *Rehm, Klawonn & Kruse (2007)* on SDS2. Based on these comparisons, we could conclude that our proposed approaches were efficient to detect the outliers in the analyzed synthetic data sets.

## EXPERIMENTS ON THE REAL DATA SETS

The performances of suggested approaches have been also tested on the real data sets, which are given in Table 5. All these data sets are the modified versions of the original data sets in UCI machine learning repository (UCIML) which have been processed for use in benchmarking for outlier detection. The descriptions of these data sets can be seen at the Outlier Detection Data Sets (ODDS) library (*Rayana, 2016*), the directory of Unsupervised Anomaly Detection of Harvard Dataverse (*Goldstein, 2015*) and UCIML (*Dua & Graff, 2017*).

The Wine data set (*wine*) in Table 6 was imported from ODDS. It is a down-sampled version of the original Wine dataset in UCIML. While the original data set contains 13 features and 1 class variable with three classes, in the modified version of it in ODDS, the class 2 and 3 have been labelled as normal data points and the class 1 has been labelled as outlier class with 10 data points.

The data sets Wisconsin breast cancer (*b-cancer*), letter recognition (*letter*), pen-based recognition of handwritten text (*pen-global*) and Statlog Landstat Satellite (*satellite*) were imported from the Harvard Dataverse repository (*Goldstein, 2015*). To use in benchmarking studies for unsupervised anomaly detection, these data sets have been generated from their original versions in UCIML.

The Wisconsin breast cancer data set in UCIML contains the records for benign and malignant cancer types. In the modified version of this data set (*b-cancer*), the malignant

Cebeci et al. (2022), *PeerJ Comput. Sci.*, DOI 10.7717/peerj-cs.1060

**Table 4  Outliers detected in the synthetic data sets by some methods in R environment.**

| Data set | Outlier detection packages | | | *Rehm, Klawonn & Kruse (2007)* |
|---|---|---|---|---|
| | **DMwR** | **mvoutlier** | **HDoutliers** | |
| SDS1 | 20 104 121 122 123 124 125 126 127 128 | 86 to130 | 121 122 123 124 125 126 127 128 129 130 | – |
| SDS2 | 2 14 33 35 40 41 42 43 44 45 | 9 14 40 41 43 44 45 | no outlies detected | 14 35 41 42 43 44 45 |

**Table 5  Real data sets used for evaluation of the proposed approaches.**

| Data set | Size | Features | Outliers (%) |
|---|---|---|---|
| b-cancer | 367 | 30 | 10 (2.70) |
| letter | 1600 | 32 | 100 (6.25) |
| pen-global | 809 | 16 | 90 (11.10) |
| satellite | 5100 | 36 | 75 (1.47) |
| wine | 129 | 13 | 10 (7.70) |

**Table 6  Number of outliers detected on the real data sets.**

| Data set | $c$ | Approach 1 | | | Approach 2 | | |
|---|---|---|---|---|---|---|---|
| | | $\alpha = 0.01$ | $\alpha = 0.025$ | $\alpha = 0.05$ | $\alpha = 0.01$ | $\alpha = 0.025$ | $\alpha = 0.05$ |
| b-cancer | 2 | 10 | 16 | 21 | 9 | 10 | 16 |
| letter | 2 | 40 | 190 | 491 | 18 | 73 | 242 |
| pen-global | 3 | 100 | 180 | 239 | 83 | 140 | 206 |
| satellite | 2 | 78 | 153 | 372 | 61 | 95 | 207 |
| wine | 2 | 2 | 7 | 16 | 2 | 3 | 8 |

class in the original data set has been downsampled to 21 points, which are considered as outliers, while points in the benign class are labelled normal data points. This data set includes totally 367 data points with the rate of 2.72% of outliers.

The original letter recognition data set in UCIML includes 16 features of 26 uppercase letters in the English alphabet. The original data set has been reorganized for outlier detection by subsampling data from three letters to form the normal class and randomly concatenate pairs of them. The *letter* data set used in this study contains 1600 data points with 32 features.

In UCIML repository, the original data set pen-based recognition of handwritten text contains the handwritten digits 0–9 by 45 different people. The processed version of this data set called "global" (*pen-global*) in ODDS has been handled by keeping only the digit 8 as the normal class and sample the 10 digits from all other classes as outliers. The *pen-global* dataset has 16 features and 809 data points (*Christy, Gandhi & Vaithyasubramanian, 2015*).

The Satellite data set (*satellite*) is a modified version of the original Statlog Landsat Satellite data set in UCIML. To create this data set, the smallest three classes have been combined to form the outlier class, while all the other classes have been as the inlier class. The *satellite* data set consists of 36 features and 5,100 data points.

In Table 6, the column with head $c$ shows the suggested cluster numbers by the fuzzy validation indices for the examined real data sets. According to the outlier detection analyses at these numbers of clusters, Approach 1 at the threshold level of 1% and Approach 2 at the threshold level of 2.5% were completely successful to find all the outliers in *b-cancer* data set. Both the proposed approaches detected more outliers than the reported number of outliers for the *letter* data set. Approach 2 with the threshold level of 2.5% produced the nearest result to the numbers of outliers in the *letter* data set. Approach 1 at the threshold level of 1% successfully detected all of the outliers marked in *pen-global* data set whereas

Approach 2 found the most of them at the same threshold level. Approach 1 with the threshold level of 1% was again successful to find the outliers in the *satellite* data set while Approach 2 detected less number of outliers for the same data set. Finally, for the *wine* data set, Approach 1 with the threshold level of 2.5% and the Approach 2 with the threshold level of 5% gave the closest number of outliers to those reported for this data set.

As a general evaluation of the results, we could conclude that when the dimensionality of data sets increases we recommend to use Approach 1 at the lower threshold levels. Working with the threshold level of 1% may practically be enough for detecting outliers in many cases. As explained in the introduction of the approaches in 'Proposed Approaches for Outlier Detection', since Approach 2 detects the most probable outliers it will return less number of outliers for the analyzed data sets.

## CONCLUSIONS

In this study, we introduced and tested two novel approaches to detect outliers using the typicality degrees obtained from unsupervised possibilistic and fuzzy clustering algorithms. Based on the experiments, the proposed approaches seem promising in detecting outliers in different kind of multidimensional data sets. Additionally, the proposed approaches are simple to implement and do not need any parameter except only a user-defined alpha value as the threshold typicality level. In this study, although we tested the proposed algorithms on the typicality degrees from UPFC algorithm, they can be applied with the typicality results from the other PCM-like algorithms. Therefore, they can be easily used for achieving considerably good performance in most of the application domains.

However, both of the proposed approaches provide the same results when the parameter $c$ is chosen close enough to the real number of clusters in data sets, Approach 2 tends to give a smaller number of outliers, usually at the higher levels of number of clusters used in the runs of the clustering algorithm. Although the outliers could be detected from the partitioning results done for the number of clusters below and above the actual number of clusters, we recommend detecting the outliers by using the partitioning result, which is suggested by the majority of fuzzy internal indices. We also recommend using Approach 1 at the lower threshold levels, *i.e.*, 1%, because it worked well for majority of the analyzed real data sets.

In this study, although the proposed approaches have been tested to detect the outliers on some well-known real data sets, we still need some further studies to examine their performances at different densities of outliers in larger data sets. The performance of outlier detection based on clustering is also closely related with the selected values of parameters which are used in the runs of clustering algorithms. Therefore, the efficiency of the proposed approaches should be examined regarding the parameters used by UPFC such as different settings of fuzziness and typicality exponents, different weights of possibilistic part of the objective function, and the different initialization techniques in generation of the prototypes used in UPFC runs.

### Funding
This study has been funded by the Unit of Scientific Research Projects of Çukurova University in Adana, Turkey (grant number FBA-2019-10285). The funders had no role in study design, data collection and analysis, decision to publish, or preparation of the manuscript.

### Grant Disclosures
The following grant information was disclosed by the authors:
Scientific Research Projects of Çukurova University in Adana, Turkey:  FBA-2019-10285.

### Competing Interests
The authors declare there are no competing interests.

### Author Contributions
- Zeynel Cebeci analyzed the data, performed the computation work, authored or reviewed drafts of the article, and approved the final draft.
- Cagatay Cebeci analyzed the data, prepared figures and/or tables, authored or reviewed drafts of the article, and approved the final draft.
- Yalcin Tahtali conceived and designed the experiments, performed the experiments, prepared figures and/or tables, authored or reviewed drafts of the article, and approved the final draft.
- Lutfi Bayyurt conceived and designed the experiments, performed the experiments, prepared figures and/or tables, authored or reviewed drafts of the article, and approved the final draft.

### Data Availability
Supplemental data and code are publicly accessible at Github: https://github.com/zcebeci/odetector.

### Supplemental Information
Supplemental information for this article can be found online at http://dx.doi.org/10.7717/peerj-cs.1060#supplemental-information.

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
