# Peer review of "Two novel outlier detection approaches based on unsupervised possibilistic and fuzzy clustering"

_PeerJ Computer Science, doi:10.7717/peerj-cs.1060_

## Round 0.1 · original submission · Major Revisions

The reviewers have found your paper valuable but they also raised some concerns. Please address these issues and prepare a revised version. Thanks.

Reviewer 1 ·

Basic reporting

The article is written in clear, professional English.
Introduction and background provided are sufficient.
Figures and tables are clearly presented.
Algorithms and equations are clearly presented.

Some Typos:
1. Line 105 – correct “not scalable well” to “not scalable”
2. Line 182 – correct to “Unsupervised Fuzzy Possibilistic Clustering algorithm by Wu et al. (2010)”
3. Line 220 – correct to “a typicality degree is a possibilistic”
4. Line 204 – UPFC has a fuzzy membership matrix which it updates. What do you mean by “UPFC also needs not a fuzzy membership matrix from a previous FCM run”?

Experimental design

Rigorous investigation, both literature and empirical can be improved, with comments below for consideration:

1. Latest literature found in related section is dated 2017. Would it be possible to include latest possibilistic clustering based outlier detection within last 3 years?
2. Line 129 states that choice of clustering algorithm for outlier detection is challenging. The motivation to apply proposed approach in Algorithm 1 to UPFC clustering and not other clustering algorithms is unclear, when there are many other possiblistic clustering algorithms.
3. It would be helpful to know more about clustering-based outlier detection, if the outlier detection approaches are explained further, in addition to explain that they are k-means, PAM or hard-clustering based outlier detection, such as in Line 139, 140. How are the outlier detection implemented in these techniques (like explained in Line 146-150)? How are they different from the proposed? What are the weaknesses of such detection techniques?
4. Line 168 to 175 are clustering algorithms extended to detect outliers. Do extended approach outperforms integrated approach details in Line 164-167?
5. Integrated methods like Treerattanapitak and Jaruskulchai (2011) performs well, it is unclear on motivation to improve extended approach and not integrated approach.
6. How does Huang et al (2017) ROCF, an extended approach like yours, perform outlier detection? What are the strengths and weaknesses of ROCF? Is it similar approach to yours? In what way are they different? Is it a parameter-free approach or otherwise? Sharing these would help readers understand the constrast and better understand the knowledge gap you are trying to fill.
7. Line 175, an existing cluster detection algorithm was explained. It is not always clear in the literature analysis whether the clustering algorithm is a sole clustering one or a clustering one to detect outliers.
8. Line 179, it is unclear why typicality degreees from possibilistic clustering algorithms cannot be directly applied to identify outlier but that it has to be fuzzy and possibilistic clustering algorithms. Kindly clarify.
9. Is UPFC parameter-free, with a and b set to 1? Can we also do this for other non-parameter free techniques? Would the proposed approach be considered parameter free if we require threshold alpha and approach 1 or 2 be set? Kindly strengthen argument of parameter-free for proposed method as opposed to others.
10. Possible to explain the effect of large c values using approach 1, wouldn’t the tendency to be an outlier increase for large number of clusters?
11. The authors explained existing techniques of setting suitable threshold alpha values for determining small clusters and they often do not handle data with large sample sizes well. It is unclear what the remedy to handling large data is, kindly explain. Instead, a remedy for setting suitable threshold value for identifying small cluster, tcs, was proposed.
12. From Line 344 to 354 and Figure 3, it is unclear whether cluster 4 or 5 are most suitable to be applied the proposed approach and when is it best to use tcs? The authors claimed that cluster 5 is successful, but this was done with human intervention and not during execution. It is unclear what happens to data points that are not outliers belonging to the small cluster or outlier cluster. It is not clear why cluster 4 with approach 1 is not the better result.
13. No comparison with existing outlier detection approaches, it is unclear how well they perform, relative to other techniques such as those in line 480 by Goldstein M, Uchida S. 2016 or how well against algorithms such as ROCF, to demonstrate in particular which real-world datasets the proposed method is highly effective compared to existing techniques. Otherwise, clarify why this was not done or is not necessary.
14. The rationale or motivation for the proposed approaches are not clear. What did the authors hope to improve by proposing approaches 1 and 2? What were the current technical limitations of similar outlier detection techniques that prompted the proposal of approach 1 and 2?
15. Is Algorithm1 executed after performing UPFC? Perhaps Line 230 can be rephrase to explain this point better.
16. Why is Algorithm 1 applied on typicality matrix and not fuzzy membership matrix? Can this rationale be explained to further highlight the motivations behind the proposal, contrasting with existing techniques?
17. It has not been demonstrated clearly whether the outliers can be detected by UPFC without Algorithm1 through experimentation of different UPFC parameters such as a and b, which has been set at 1 in this work. Comparison of outliers identified by UPFC and UPFC with Algorithm 1 will help demonstrate the effectiveness of Algorithm 1. Will different values of a and b help improve UPFC clustering and thus outlier detection?
18. It is unclear if the proposed method can be applied to and be effective for identifying outliers, with other fuzzy possibilitistic clustering algorithms or other clustering algorithms, as the typicality matrix can be constructed from any clustering results. Can discussion on this be included?
19. It is unclear how proposed method decide on the cluster to use
20. Algorithm 1, Line 16: Correct to “for i = 1 to c do” - missing 1
21. Algorithm 1, Line 27: Correct to “T[i , k]” - comma
22. Algorithm 1, Line 31: Correct to “isOutlier” - missing i in outlier

Validity of the findings

1. The authors' algorithms for approach 1 and 2 can be replicated, with the parameters applied presented.

2. It will be helpful for replication if all modified datasets are shared.

3. Conclusions are well-stated and link to research question and supporting results.

Additional comments

More literature analysis on rationale, in-depth contrast of technical details of existing will help readers appreciate contribution of this work, and identify technical differentiation.

Further discussion of application of the proposed methods with other clustering algorithms, with or without empirical output, will help other practitioners or interested practitioners and their competitiveness with other existing techniques, will demonstrate applicability, impact and potential of proposed methods.

·

Basic reporting

Thank you for the opportunity of reviewing this manuscript. I think that the popularity of outlier detection is increasing as the number of potential applications increases. The manuscript is generally well written, but I have some concerns (listed below) that should be addressed to improve the quality of the paper.

At line 189 the manuscript proposed that the fuzzy membership degree to each data point is based on their distances to the cluster centers. On line 97, reviewing Cebeci (2020), the paper states that "In outlier detection, the overall performance of a proximity-based method can be easily affected by the metrics used with it". Therefore, the manuscript should explain why it isn't problematic to use a measure of distance from the cluster centers to evaluate the "outlierness" of each point.

Experimental design

It is not clear how the threshold level alpha for Approach 1 and 2 fits the unsupervised method. For the artificial datasets, alpha was set to 0.01 and 0.05 respectively (line 230 and 255). Although, for Approach 2, it looks like alpha can go up to 0.10 (10%). With real datasets, the value of 0.5 and 0.10 for Approach 2 does not work very well. Instead, the manuscript proposes 0.025 as the optimal value. How this unstable behavior of alpha fits the unsupervised detection method?

Validity of the findings

Last but not least, I believe that to understand the relative performance of the proposed methods, it would be advisable to provide benchmarks that compare the performance of existing (best) methods (maybe both KNN and density/distance-based) with the performance of the proposed methods. Without benchmarking, the reader is left wondering if these proposed methods are an improvement of existing standards and (if) how they can fit the practical business application. The benchmarking will provide evidence of the validity of your proposed methods.

---

## Round 0.2 · accepted · Accept

I have looked at the authors' response to reviewer 2's concerns. I am satisfied that they have been addressed. The paper can be accepted. Congratulations

Reviewer 1 ·

Basic reporting

I thank the authors for clarifying all comments. No other comment.

Experimental design

I thank the authors for clarifying all comments and show further comparative analysis with other existing techniques. No other comment.

Validity of the findings

I thank the authors for sharing their code and data. This will be helpful to other researchers. No other comment.

Additional comments

Overall, the authors have addressed the comments clearly. I have no further comment.